[Reviews · NeurIPS 2017]

Reviewer 1



This paper presents an interesting algorithm to recover 2.5 and 3D shapes from 2D images. The paper is well written and very clear. I have only a few concerns: 1- I think that reconstruction can be only done from a monocular view up to scale. In the proposed algorithm I think that this issue can be overcome by the use of shape prior information. I think the authors should comment this issue in the problem 2- Related to the previous point, the success of the algorithm seems to depend deeply on the object prior information. So one of the main limitations of the usability of the algorithm is that it only works for specific trained objects classes. If this is the case it should be mentioned in the paper 3- I think that it should also be interesting a motivation paragraph. Who should be interested in this? (besides the academic interest), why this approach is more interesting than stereo, depth cameras..... I think that an interesting object not tackled in the paper are human shapes and faces. Can this technology be usable in this cases?

Reviewer 2



This paper describes a technique for estimating voxelized 3D shapes from single images. As directly predicting 3D shapes from images is hard, the paper proposes to separate the problem into 2 tasks - inspired by Marr's theory about vision. In the first part the method takes the image as input and predicts several intrinsic quantities ("2.5D sketch"), in particular depth, surface normals and the 2D silhouette using an encoder-decoder architecture. This information is fed into a second encoder-decoder network which predicts the volumetric 3D representation at 128^3 resolution. The advantage of the method is that the second part can be trained on synthetic data allowing for "domain transfer" based on the 2.5D sketches which are invariant to the actual appearance. Experiments (mainly qualitative) on Shapenet, IKEA and Pascal3D+ are presented. Overall, I liked the idea. The paper is clear, well motivated and written and the qualitative results seem convincing. However, I have some concerns about the fairness of the evaluation and the self-supervised part which I like to have answered in the rebuttal before I turn towards a more positive score for this paper. Positive aspects: + Very well written and motivated + Relevant topic in computer vision + Interesting idea of using intermediate intrinsic representations to facilitate the task + Domain transfer idea is convincing to me (though the self-supervision and fine-tuning of the real->sketch network are not fully clear to me, see below) + Qualitative results clearly superior than very recent baselines Negative aspects: - Self-supervision: It is not clear to me why this self-supervision should work. The paper says that the method fine-tunes on single images, but if the parameters are optimized for a single image, couldn't the network diverge to predict a different shape encapsulated in the shape space for an image depicting another object while still reducing the loss? Also, it is not very well specified what is finetuned exactly. It reads as if it was only the encoder of the second part. But how can the pre-trained first part then adapted to a new domain? If the first part fails, the second will as well. - Domain adaptation: I agree that it is easy to replace the first part, but I don't see how this can be trained in the absence of real->sketch training pairs. To me it feels as the method profits from the fact that for cars and chairs pretraining on synthetic data already yields good models that work well with real data. I would expect that for stronger domain differences the method would not be able to work without sketch supervision. Isn't the major problem there to predict good sketches from little examples? - Baselines: It is not clear if the direct (1-step) baseline in the experiments uses the same architecture or at least same #parameters as the proposed technique to have a fair evaluation. I propose an experiment where the same architecture is used but insteads of forcing the model (ie, regularizing it) to capture 2D sketches as intermediate representations, let it learn all parameters from a random initialization end-to-end. This would be a fair baseline and it would be interesting to see which representation emerges. Another fair experiment would be one which also has 2 encoder-decoder networks and the same number of parameters but maybe a different distribution of feature maps across layers. Finanlly a single encoder-decoder architecture with a similar architecture as the proposed one but increased by the number of parameters freed by removing the first part would be valuable. From the description in the paper it is totally unclear what the baseline in Fig. 4. is. I suggest such baselines for all experiments. - DRC: It should be clarified how DRC is used here and if the architecture and resolution is adopted for fairness. Also, DRC is presented to allow for multi-view supervision which is not done here, so this should be commented upon. Further the results of the 3D GAN seem much more noisy compared to the original paper. Is there maybe a problem with the training? - Quantitative Experiments: I do not fully agree with the authors that quantitative experiments are not useful. As the paper also evaluates on ShapeNet, quantitative experiments would be easily possible, and masks around the surfaces could be used to emphasize thin structures in the metric etc. I suggest to add this at least for ShapeNet. Minor comments: - Sec. 3.1: the deconvolution/upconvolution operations are not mentioned - It is unclear how 128^3 voxel resolution can be achieved which typically doesn't fit into memory for reasonable batch sizes. Also, what is the batch size and other hyperparameters for training? - Fig. 8,9,10: I suggest to show the predicted sketches also for the real examples for the above mentioned reasons.

Reviewer 3



This paper has potential but is currently written in a confusing way. For example the paper claims the model is self-supervised in several places. This makes no sense. The model is trained using full supervision (3d models, 2.5 depths, normals and segmentations). What happens is that at test time reconstruction is achieved through optimization. Here's how it could have been better written: There are two neural networks, a) predicts object 2.5D depth, normals and foreground-background segmentation, b) predicts a voxel-based 3D reconstruction of the object from the 3 outputs of the first neural network. The networks are trained independently. At test time, reconstruction is computed through gradient descent over the weights of the voxel-net's encoder, with the rest of the model parameters fixed. This perspective sheds light on merits and limitations of the approach. Limitations first: it must be very slow (how slow?), it isn't trained for the task at hand, e.g. the model isn't trained to optimize the reconstruction. Merits: the idea of having a fixed voxel-net decoder is interesting and seems to produce good results. The idea of the 2.5D net, which can more easily be trained on real data, as setting constraints on the voxel-net is also interesting. The technical section begins with a hand-waving explanation of the architectures. How are the 3 inputs to the voxel net combined ? This basic information should be in the paper, so it is minimally self-contained. One of the most important parts of the paper is fit into 10 lines (section 3.4) without much discussion or details, making it hard on the reader. The rebuttal addressed some of my concerns and I think it can be accepted.